# Study of changes in brain dynamics during sleep cycles in dogs under effect of trazodone

**Magaly Catanzariti[1,2], Alejandra Mondino[3], Pablo Torterolo[4], Hugo Aimar[1,2], Natasha J. Olby[3]\*, Diego M. Mateos** [1,2,5,6]\*

**1** Consejo Nacional de Investigaciones Científicas y Técnicas (CONICET), Buenos Aires, Argentina, **2** Instituto de Matemática Aplicada del Litoral (IMAL-CONICET-UNL), CCT CONICET, Santa Fé, Argentina, **3** Department of Clinical Sciences, North Carolina State University, Raleigh, North Carolina, United States of America, **4** Laboratory of Sleep Neurobiology, Department of Physiology, Facultad de Medicina, Universidad de la República, Montevideo, Uruguay, **5** Facultad de Ciencia y Tecnología, Universidad Autónoma de Entre Ríos (UADER), Oro Verde, Entre Ríos, Argentina, **6** Achucarro Basque Center For Neuroscience, Leioa, Vizcaya, Spain

\* njolby@ncsu.edu (NO); mateosdiego@gmail.com (DMM)

## Abstract

Sleep is a fundamental biological process essential for cognitive function, memory consolidation, and overall health in both humans and animals. Dogs, in particular, share many physiological and neurological similarities with humans, making them a valuable model for sleep research. Similar to humans, dogs can experience sleep disorders that disrupt sleep cycles and impair cognitive function. While serotonin antagonist and reuptake inhibitors (SARIs), like trazodone, have been shown to alleviate these conditions in dogs, their underlying neural mechanisms remain poorly understood. This study investigates the effects of trazodone on brain dynamics in healthy dogs using electroencephalographic (EEG) analysis. We compared treated subjects with a control group by characterizing EEG activity across wakefulness, drowsiness, Non-REM (NREM), and REM sleep states. Hypnogram analysis was used to assess sleep architecture, including alterations in cycle patterns and time spent in each stage. Additionally, we examined linear and non-linear EEG dynamics using Power Spectral Density (PSD), Permutation Entropy (PE), and Lempel-Ziv complexity (LZC), as well as connectivity changes through Phase Lag Index (PLI) and coherence analysis. Our findings indicate that trazodone significantly alters sleep structure by modifying sleep cycles, reducing power in lower frequency bands across most sleep stages, and increasing power in frequencies above 13 Hz during Drowsiness and NREM. Furthermore, treated dogs exhibited increased signal entropy and complexity in lower frequency bands across all sleep stages, along with a reduction in brain connectivity in most stages and frequency bands. These results provide new insights into the short-term effects of trazodone on brain activity during sleep, with potential implications for its clinical use as a sleep aid in dogs.

**Data availability statement:** https://zenodo.org/records/17062482.

**Funding:** This research was supported by the following grants: NO: The Dr. Kady M. Gjessing and Rhanna M. Davidson Distinguished Chair of gerontology. HA: # MinCyT-FonCyT PICT-2019 N° 01750 PMO BID; grant CONICET-PUE-IMAL # 229 201801 00041 CO; grant CONICET-PIP-2021-2023-GI #11220200101940CO and grant UNL-CAI+D #50620190100070LI. PT:CSIC-I+D groups 2022- group ID-22620220100148 - Uruguay. The funders had no role in study design, data collection and analysis, decision to publish, or preparation of the manuscript.

**Competing interests:** The authors have declared that no competing interests exist.

## Introduction

Sleep is a vital physiological process that has played a crucial role since the earliest stages of animal evolution [1]. Despite ongoing research, the precise role of sleep remains elusive. However, it is evident that sleep is crucial for various functions, including energy restoration, physical recovery, and growth, particularly in young animals [2]. Additionally, sleep is essential for the consolidation of learning and memory acquired during wakefulness [3], significantly contributing to memory enhancement by facilitating the transfer of information from short-term to long-term memory storage [4]. Numerous studies highlight the negative effects of sleep deprivation on both physical [5,6] and cognitive performance [7,8]. These effects manifest as reduced reaction times and impaired decision-making, increasing the likelihood of accidents and errors [8,9]. Therefore, investigating brain dynamics during sleep cycles is paramount to comprehending both basic and complex behavioral functions. Understanding these dynamics holds promise for unraveling the intricate mechanisms underlying sleep and its impact on overall well-being.

To gain a deeper understanding of these brain dynamics, the gold standard technique is polysomnography (PSG), which involves the simultaneous recording of electroencephalogram (EEG), electromyogram (EMG), and electrooculogram (EOG) [10–12]. PSG can assess both the quantity and quality of sleep. Additionally, the analysis of EEG signals using various mathematical tools, such as Power Spectral Density (rPSD), Lempel Ziv Complexity, and Permutation Entropy, allows the quantification of brain dynamics during different sleep stages and their alterations by pathologies and drugs [13–15].

Extending this research to animal models, the dog has been proposed as an appropriate model for sleep research due to its sleep cycles being synchronized with those of humans. Moreover, domestic dogs are susceptible to comparable sleep disorders such as sleep apnea, narcolepsy, and REM sleep disorder [16,17]. Studies on dogs have shown analogous patterns of reduced performance and impaired physiological function due to sleep deprivation, mirroring disturbances in sleep architecture and respiration observed in obstructive sleep apnea and chronic painful conditions such as osteoarthritis [16,18,19]. Ageing in dogs also leads to changes in sleep, with older dogs exhibiting nocturnal sleep fragmentation and increased daytime sleep episodes [20]. Similar to humans, a non-invasive polysomnography technique has been developed and validated for this species [21].

Insomnia is one of the most prevalent and significant sleep disorder. Previous research has demonstrated the efficacy of certain drugs in the treatment of insomnia in humans. In various clinical contexts, hypnotic drugs are often recommended as a viable treatment option for sleep disorders [22–24]. One such drug, trazodone, is a triazolopyridine derivative with antidepressant, anxiolytic, and hypnotic properties [25]. It has been shown to effectively improve sleep quality in human patients with sleep disorders and even in those with Alzheimer's disease [26–29]. In veterinary medicine, studies have shown that trazodone has positive effects on dogs, including reducing stress in hospitalized dogs [30], alleviating anxiety [31], and aiding in postoperative recovery [32]. For further evidence on the use of trazodone in dogs,

see [33]. However, as far as we know, there is a lack of literature documenting the qualitative and quantitative changes induced by trazodone at the level of neuronal network dynamics. Therefore, this study both qualitative and quantitative examines dogs' EEG recordings to investigate the neuronal patterns altered by trazodone.

Our hypothesis posits that trazodone causes modifications in brain dynamics in dogs during different sleep states. To explore this, we studied dogs' EEG recordings using four methodologies: (i) Sleep State Duration and Changes in the Sleep Cycle, (ii) Power Spectral Density (PSD) analysis, (iii) non-linearity analysis using Lempel Ziv Complexity and Permutation Entropy, and (iv) connectivity studies based on Phase Lag Index (PLI) and coherence analysis. Through these analyses, we observed that animals receiving trazodone experienced a) changes in the structure of the sleep cycle, b) an increase in power within major bands above 10 Hz in NREM states, c) increased signal complexity and entropy during wakefullness, drowsiness and NREM sleep in the lower frequency bands, and d) modifications in brain connectivity in most EEG bands.

These results provide new insights into the short-term effects of trazodone on brain activity during sleep. By enhancing our understanding of trazodone's short-term influence on neural dynamics during sleep, this study paves the way for future investigations aimed at optimizing the therapeutic use of trazodone in managing sleep disorders in dogs, and potentially extending these insights to other species, including humans.

## Materials and methods

### Population study

Twelve young, healthy domestic dogs (mean aged $60.8 \pm 35.16$ month), representing a mix of breeds and sexes, participated in this study (mean weight $20.08 \pm 9.1$ Kg). All animals underwent three stages carried out on different days: acclimatization, recording with trazodone, and recording without trazodone. A summary of the information for each participant is in Table 1. All procedures were approved by the Institutional Animal Care and Use Committee at North Carolina State University (NCSU; protocol number 21-303). The dogs were client-owned and were serving as age control in an ongoing longitudinal study on neuro-aging at the NCSU College of Veterinary Medicine, as described in previous publications [34–36]. Before participation, informed consent was obtained from all dog owners, who reviewed and signed consent forms.

### Polysomnographic studies

The dogs underwent polysomnographic recordings, during which electroencephalogram (EEG), electrooculogram (EOG), electromyogram (EMG), and electrocardiogram (ECG) signals were simultaneously collected, following a slightly modified version of the protocol by [37]. As shown in Fig 1A, four active EEG electrodes were placed: F3 and F4 (left and

**Table 1**. Information on the population of dogs used in the study.

| Name | Breed | Sex | Weight (kg) | Age (months) |
|------|-------|-----|-------------|--------------|
| 1 | Labrador Retriever | FS | 27,1 | 103,8 |
| 2 | Shitzu | MC | 6,2 | 83,2 |
| 3 | Beagle | MC | 7,5 | 15,7 |
| 4 | Pitbull mix | MC | 22,2 | 60,4 |
| 5 | Mix breed | FS | 21,2 | 35,7 |
| 6 | Mix breed | MC | 14,5 | 152 |
| 7 | Pitbull mix | MC | 30,6 | 45,2 |
| 8 | Pitbull | FS | 23,1 | 36,1 |
| 9 | Lab mix | FS | 21,4 | 57,5 |
| 10 | Maltese | F | 5,4 | 79,6 |
| 11 | Pitbull | MC | 30,3 | 48 |
| 12 | Labrador Retriever | MC | 31,5 | 46,1 |

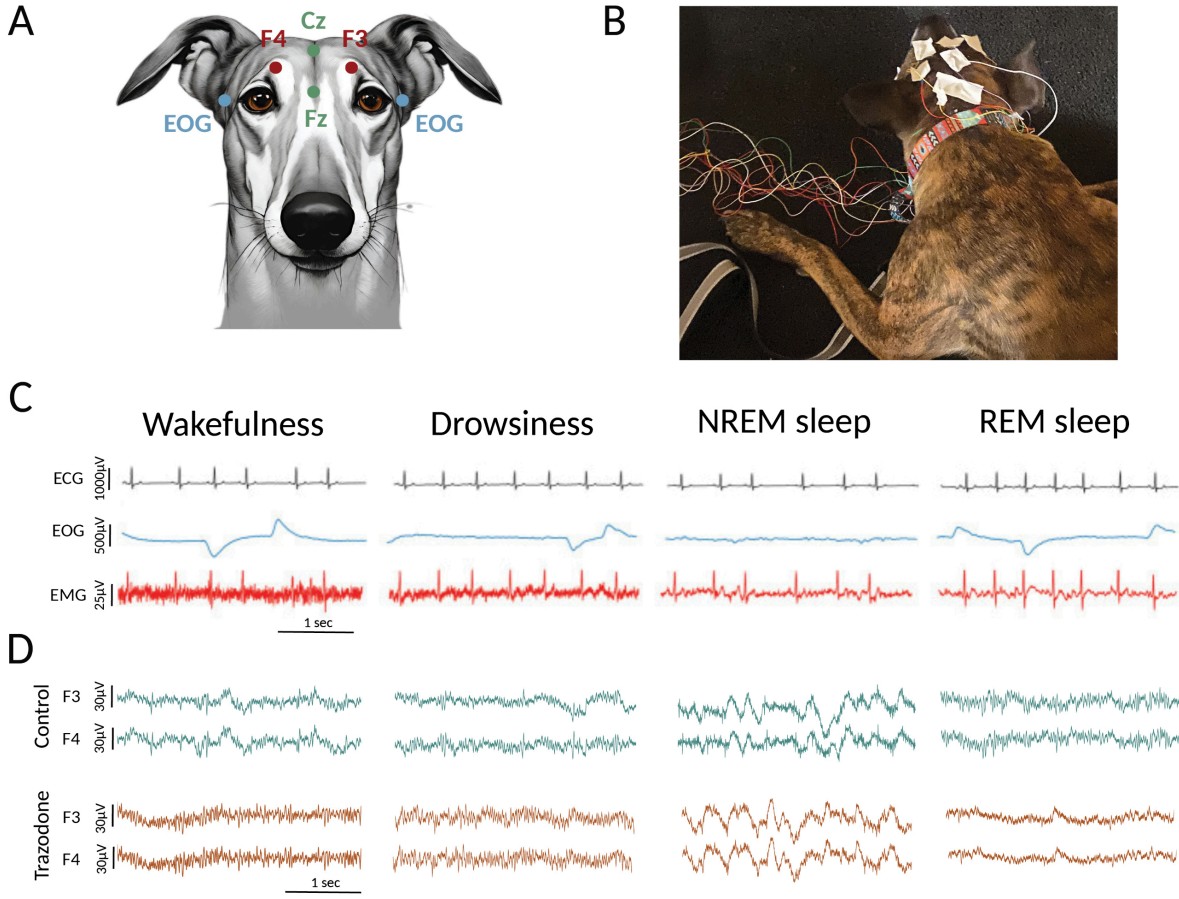

**Fig 1. Illustration of polysomnographic recordings conducted in dogs. (A)** Electrode placements included four active EEG electrodes positioned over the midline, left and right frontal regions (Fz, F3, and F4, respectively), and the vertex (Cz). A reference electrode was placed over the external occipital protuberance (Oz). Bipolar EOG signals were recorded using electrodes placed on the left and right zygomatic arches near the lateral canthus of each eye (blue dots). A ground electrode was positioned on the left temporal muscle, EMG and ECG recordings were also performed but are not shown in the figure. **(B)** Photograph of a dog undergoing polysomnographic recording while resting on its bed. **(C)** Representative signals from the polysomnographic study, including electrocardiogram (ECG), electrooculogram (EOG), and electromyogram (EMG), illustrating characteristic patterns during wakefulness, drowsiness, non-rapid eye movement (NREM) sleep, and rapid eye movement (REM) sleep in control conditions. **(D)** Two representative EEG recordings (channels F3 and F4) for a dog under control conditions and after trazodone administration, shown across the different behavioral states.

right frontal, respectively), Fz (midline frontal), and Cz (vertex). This electrode placement was designed to capture activity in the frontal and parietal cortices. The EEG electrodes were referenced to an Oz electrode located over the external occipital protuberance.

Bipolar EOG signals were recorded with electrodes placed on the left and right zygomatic arches, near the lateral canthus of each eye. EMG signals were recorded bipolarly from two electrodes placed on the dorsal neck muscles bilaterally. Additionally, ECG signals were captured with an electrode positioned over the fifth intercostal space, referenced to the Cz electrode. All recordings were conducted using gold-coated electrodes (Genuine Grass 10 mm Gold Cup, Natus Medical Inc), which were secured with SAC2 electrode cream (Cadwell Laboratories) after preparing the skin with a skin preparation solution and Signa Spray electrode solution (Parker Laboratories). A ground electrode was positioned over the left temporal muscles. The polysomnographic recordings were managed using Cadwell Easy II software (Cadwell Laboratories). Before each recording session, electrode impedance was verified to ensure it remained below 20 $k\Omega$, and

a sampling rate of 400 Hz was maintained throughout the study. Recordings were conducted in a quiet, dimly lit room with white noise generated from a laptop, and the temperature was maintained at 20°C. Owners brought their dogs for polysomnography on three separate days: an adaptation day, intended to minimize the "first-night effec" [37], and two different recording days (with and without trazodone). For added comfort, owners were encouraged to bring familiar bedding items, such as their dogs' own beds or blankets. Fig 1B illustrates a dog undergoing a polysomnography session.

On the adaptation day, a 30-minute recording session was conducted to allow the dogs to acclimate to the room and the recording setup. Subsequently, two additional sessions, in differents days, were performed to record data under two conditions: (i) without medication and (ii) with trazodone. The order of these conditions (with or without trazodone) was counterbalanced. The drug dose used was 5 mg/Kg oral route, which is the dose used as a sedative in dogs [30]. Following the adaptation session, the initial condition (with or without the drug) was randomly assigned, and during the third visit, the opposite condition was implemented. The interval between the adaptation and recording days was kept under two weeks. During the testing day, a 2-hour polysomnography session was conducted, beginning between 12:30 PM and 1:30 PM to align with the dogs' natural nap times. If any dog displayed signs of anxiety or removed its electrodes, the session was paused.

## Signal preprocessing

The first step was preprocessing the signals, applying a bandpass filter between [1,70] Hz, along with a notch filter at 60 Hz to eliminate power line noise. Sleep states, drowsiness, NREM sleep, and REM sleep—were were manually scored in 3-second epochs, following established practice in canine and others animals polysomnography [12,21,37]. Short epochs were chosen for two reasons: (i) dogs were unsedated and often produced movement or muscle artifacts, which are easier to detect and exclude in shorter segments; and (ii) transitions between vigilance states occur more rapidly in dogs than in humans, so finer temporal resolution improves identification of NREM and REM states.

Sleep states were stratified by expert scorers according to the defining features of the polysomnographic recordings. Particularly, wakefulness was characterized by fast EEG activity, high EMG levels, and frequent eye movements. drowsiness presented with EEG activity similar to wakefulness but with reduced muscle tone and lower frequency of eye movements. NREM sleep was identified by low-frequency, high-amplitude oscillations, primarily in the delta band, along with decreased muscle activity and occasional eye movements. REM sleep was characterized by high-frequency EEG activity, minimal muscle tone, and frequent rapid eye movements, which appeared as artifacts in the EEG. Fig 1C illustrates examples of ECG, EOG, and EMG signals observed in each of these states. Fig 1D shows the EEG in control and trazodone conditions across behavioral states.

For the quantitative analyses, all epochs extracted to each behavioral state were included. In contrast, for the qualitative analyses of the EEG (PSD, non-linearity and connectivity), only artifact-free epochs were selected. On average, the proportion of artifact-free epochs varied across sleep stages: under control conditions, 7.7% of wakefulness, 41.1% of drowsiness, 33.5% of REM, and 100% of NREM epochs were retained. Following trazodone administration, these values increased to 17.7%, 58.9%, 53.1%, and 100%, respectively (S1 and S2). A summary of the time spent in each behavioral state for every dog under control and trazodone conditions can be seen in (S3). To mitigate potential bias from unequal artifact rates, quantitative EEG analyses were performed using five artifact-free epochs per state per dog.

## Hypnogram quantification

We focused on studying and quantifying the hypnograms of animals with and without trazodone. Fig 2A shows representative hypnograms from a dog under both conditions.

We quantified the hypnograms using three analyses. First, we measured the time spent in the different sleep states by individuals in both groups. For each animal, we calculated the number of epochs in every sleep state, divided by the total number of recorded epochs, and multiplied by 100 to obtain the percentage of time spent in each state.

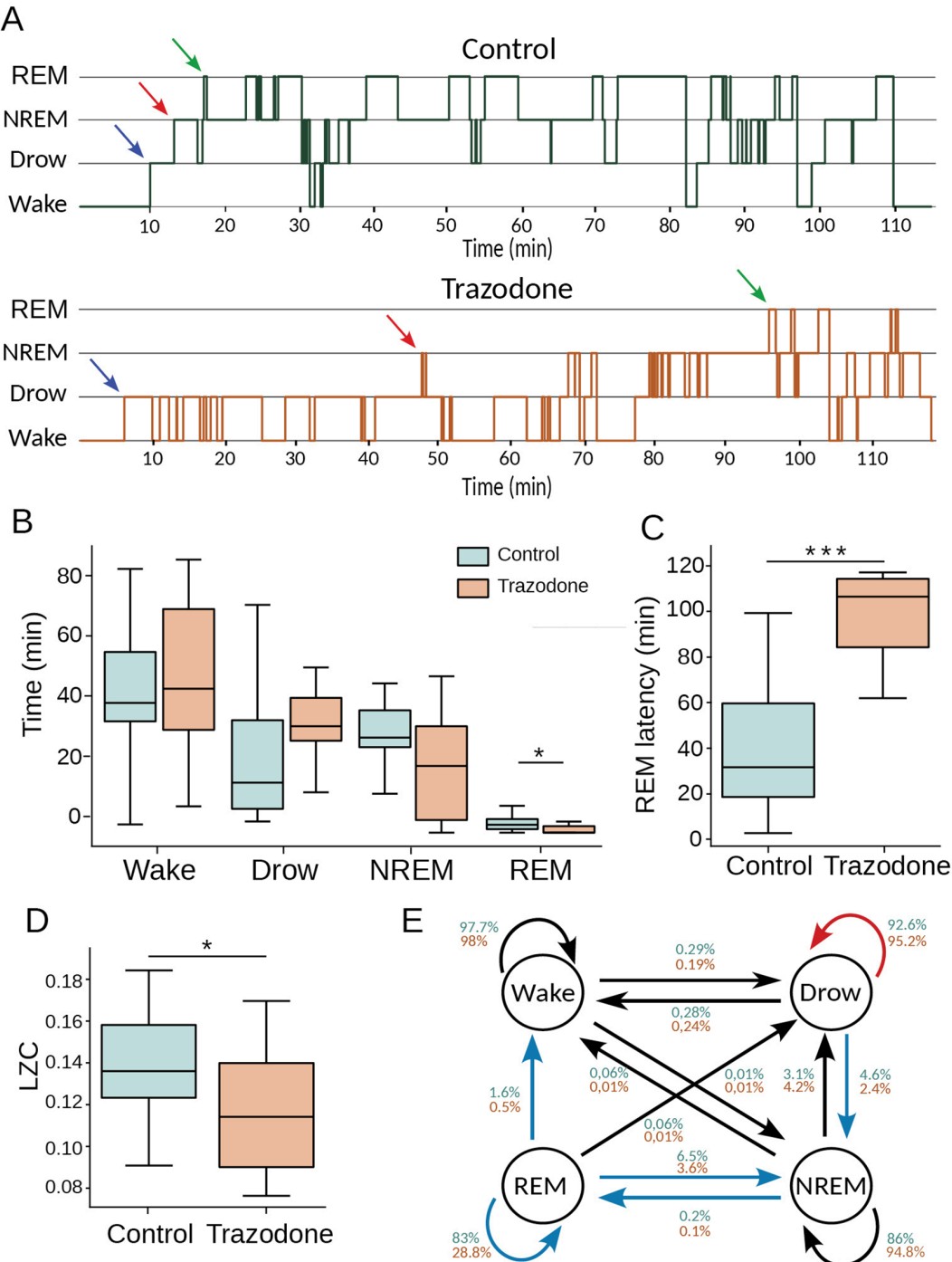

**Fig 2**. **(A) Example of a hypnogram for the same dog under control and trazodone conditions.** Arrows indicate the first occurrence of Drowsiness (blue), NREM (red), and REM (green) stages. **(B)** Time (in minutes) spent in each sleep stage under control and trazodone conditions ($*p = 0.037$). **(C)** REM sleep latency (in minutes) for both conditions ($***p = 0.004$). **(D)** Lempel-Ziv Complexity (LZC) of hypnogram sequences under both conditions ($*p = 0.023$). **(E)** Transition probability between sleep stages for control (green) and trazodone (orange) conditions. Blue arrows indicate a significant decrease, and red arrows an increase in transition probability under trazodone. Missing arrows indicate no observed transitions between stages. For all analyses, the Wilcoxon signed-rank test was used as the statistical test.

In the second analysis, we examined the dynamics of sleep cycles by assessing the complexity of the hypnograms in both groups. To quantify the difference between hypnograms structure, we applied Lempel-Ziv complexity (LZC), detailed in the Method's in the following section.

Finally, we focused on examining the probability of transitions between different sleep states throughout the entire recording. Using the hypnogram sequence, we studied the probability of remaining in or transitioning to a different sleep state in each epoch.

### EEG analysis

To standardize the analysis and mitigate bias from the variable number of available epochs, we randomly selected five artifact-free epochs per behavioral state for each dog. Each epoch had a fixed length of $N_{point} = 1200$ data points. A representative value for each state and dog was then obtained by computing the mean across the five epochs, ensuring uniformity in the dataset and facilitating robust comparisons between states.

**Power spectrum density.** The power spectrum density (PSD) quantifies signal power across frequencies. However, absolute power is confounded by inter-individual anatomical differences (e.g., skull thickness, electrode placement). To mitigate this, we computed relative power spectral density (rPSD) by normalizing the power at each frequency by the total power –in our case the [1, 50] Hz band for each dog. This normalization minimizes inter-individual differences in skull thickness or head size and allows comparisons to focus on spectral distribution rather than absolute amplitude. All the analysis was made using `Welch` function from the `SciPy` signal package in Python.

**Non-linear analysis.** Due to the significant nonlinearity of EEG signals [38], it is crucial to use specific mathematical tools that can capture the nonlinear dynamics inherent in these signals. Tools based on information theory have proven invaluable in the study of many electrophysiological signals [13,15,39,40]. In this work, we applied two widely used tools in this area to EEG signals: *Lempel-Ziv complexity* and *Permutation Entropy*. Previous studies have observed differences in the nonlinear dynamics of signals associated with various sleep states, depending on whether low or high frequencies were analyzed [41]. Due to these findings, we analyzed the EEG signals in two different frequency ranges, low frequency range from [1, 16] Hz and High frequency range from [17, 50] Hz.

**Lempel Ziv Complexity.** (LZC) is a computational method used to quantify the complexity or randomness within a sequence, such as a time series or a symbolic string. LZC examines how frequently new patterns, or distinct subsequences emerge as the sequence progresses. In essence, it divides the sequence into substrings and tracks the appearance of unique patterns. A higher LZC value indicates greater pattern diversity, reflecting increased unpredictability and complexity within the data. This approach, initially proposed by Lempel and Ziv in 1976 [42], provides valuable insights into the structural richness and variability of a sequence.

LZC was applied for the analysis of both hypnograms and EEG recordings. For the hypnogram analysis, sleep states were represented as sequences of four distinct symbols: wakefulness (0), drowsiness (1), NREM (2), and REM (3). Each symbol corresponded to a specific sleep state, forming sequences such as 00001100110011222111133333... . The Lempel-Ziv Complexity (LZC) was then calculated for each sequence, providing a complexity value for each animal.

For EEG signal analysis, we first discretized the contious signal into sequences of 0 and 1s. To achieve this, we defined a threshold based on the median of the EEG signal: values above the threshold were assigned a value of 1, and those below were assigned 0, resulting in a binary sequence.

In both cases hypnogram and EEG binary sequence we applied the LZC algorithm develop by Kaspar and Schuster [43] implemented in Python.

**Permutation entropy.** (PE) is a complexity measure used to assess the unpredictability within a time series, originally introduced by Bandt and Pompe (2002). PE examines the ordering of data points within short segments of the series, uncovering structural information about the signal [44]. It works by defining ordinal patterns based on the relative order of neighboring values, capturing temporal trends such as increases or decreases. Ordinal patterns are calculated based on

two parameters: $D$, the embedding dimension, which is the number of neighboring points considered, and $\tau$, the delay, representing the number of time steps (or lag) between consecutive elements of the embedding vector. Once these patterns are identified, a probability distribution of the patterns is computed. Finally, the Shannon entropy of this distribution is calculated, providing a measure of complexity that is both computationally efficient and robust to noise, making it suitable for non-stationary data. For a more in-depth understanding of PE, see [45]. In our study, we applied PE to EEG signals and compared the results to those from Lempel-Ziv complexity analysis. For PE implementation, we used the Python package Ordpy [46].

**Connectivity analysis.** To complement our analysis, we conducted two connectivity studies to examine interactions between the different brain areas involved in the research. The first analysis employed the *Phase Lag Index* (PLI), a metric based on the phase relationship between two signals. The second analysis used the *coherence* metric, which relies on the spectral density shared between the two signals.

**Phase Lag Index.** (PLI) measure assesses the consistency of phase differences between signals over time, providing an indication of the degree of synchronization[47]. The PLI is calculated by examining the distribution of phase differences and determining the extent to which these differences are non-uniform. A PLI value close to 1 indicates strong phase synchronization, while a value close to 0 indicates no synchronization.

Since the *PLI* is based on obtaining the phase of the wave by applying the Hilbert transform, it is necessary to analyzethis metric in signals filtered into narrow frequency bands [48]. We have analyzed the most physiologically representative EEG bands: delta [2, 4] Hz, theta [5, 7] Hz, alpha [9, 11] Hz, beta [19, 21] Hz, and gamma [34, 36] Hz.

PLI analysis was performed between all pairs of channels, across all bands mentioned, and over all epochs corresponding to each sleep state of each dog. This gave us a set of 6 bands $\times$ 5 epochs $\times$ 4 states per animal. The average of the connectivity matrices for all epochs was calculated. For each dog, we obtained a 6 set of $4 \times 4$ connectivity matrices (one for each band and sleep state). To observe connectivity differences between the trazodone and basal, we calculated the difference in PLIvalues for each dog, with and without trazodone, and then averaged these differences across all animals as follows: $\Delta PLI = < PLI_{cont} - PLI_{traz} >_{N_{dog}}$. To assess the significance of these differences, we performed a surrogate analysis as follows: for each dog in both conditions, we generated surrogate EEG signals (repeated $N_s = 100$ times) and calculated the PLI for each surrogate. We then computed the mean PLI matrices for the surrogates in each condition ($PLI^s_{cont}$, $PLI^s_{traz}$). Next, we subtracted the surrogate *PLI* matrices and calculated the mean and standard deviation across all animals ($\Delta PLI^s$, $\sigma PLI^s$). A significance threshold matrix was defined as $Th = \Delta PLI^s + \sigma PLI^s$. Finally, we compared the $\Delta PLI$ matrix with the $Th$ matrix, visualizing only those connections that satisfies $\Delta PLI > Th$. PLI analysis was implemented on Python.

**Coherence.** analysis was used to evaluate functional connectivity between EEG signals across different channels. Coherence measures the degree of linear synchronization between two signals based on their phase and amplitude relationships [49]. Coherence values were computed using Welch's method. The analysis was performed in Python using the function `scipy.signal.coherence`, with a Hann window of 800 ms and 50% overlap. Coherence was analyzed as a function of frequency within the range of [1,50] Hz for all channel pairs. The mean coherence was calculated across all epochs corresponding to the same participant and experimental condition.

## Statistical analysis

For the quantitative of hypnograph, paired Wilcoxon signed-rank tests (two-sided) were applied. For the EEG signals analyses in wakefulness and drowsiness, paired Wilcoxon signed-rank tests (two-sided) were used. For REM and NREM, where some trazodone sessions lacked these states and two dogs lacked NREM entirely, unpaired Wilcoxon rank-sum tests (Mann–Whitney U test) were applied for state-wise comparisons. Normality was assessed using the Kolmogorov–Smirnov test. Multiple comparisons within the complexity and entropy analyses were controlled with Kruskal–Wallis tests followed by Dunn's post-hoc correction. Statistical significance was set at $p < 0.05$.

## Results

### Temporal duration and dynamics of sleep state cycles

The first analysis focused on examining the sleep cycles of the animals under control conditions and under the effects of trazodone. Fig 2A shows representative hypnograms for one dog in both conditions.

Fig 2B compares the time spent in each sleep state between the two groups. Dogs treated with trazodone did not show significant differences in wakefulness, drowsiness, or NREM sleep. However, a notable finding is observed in the REM sleep state: while all dogs in the control condition entered REM sleep, 60% of the dogs under the effects of trazodone did not. This resulted in a significant difference between the two groups in the amount of time spent in REM sleep.

A latency study was conducted for the sleep states drowsiness, REM, and NREM. This analysis measures the time it takes for the animal to enter the first sleep stage under examination (arrows in Fig 2A). For dogs that did not enter the REM state, a latency of 120 minutes (the total recording duration) was assumed. The results indicate that significant differences between groups were observed only in the REM state ($p = 0.004$), as shown in Fig 2C. Additionally, a correlation analysis was performed between REM latency, weight, and age (S5), but no significant associations were found between these variables.

Fig 2D exhibits the result of the analysis of the LZC over the hypnogram. This figure shown a significant decrease in the LZ complexity in animals treated with trazodone. This suggests that dogs on trazodone experience less variability in their waking-sleep cycles and remain in each behavioral state for longer than controls —as we can see visually in the representative hypnograms of Fig 2A.

Finally, Fig 2E, presents the transition probability matrix for sleep stage transitions between consecutive epochs ($P(epoch_t|epoch_{t+1})$) under both control and trazodone conditions. Overall, the trazodone group exhibited a general decrease in transition probabilities between sleep stages. Notably, significant reductions were observed in the transitions from drowsiness → NREM ($p = 0.03$), NREM → REM ($p = 0.002$), REM → wakefulness ($p = 0.035$), REM → NREM ($p = 0.038$), and REM → REM ($p = 0.006$). The only transition showing an increased probability under trazodone was drowsiness → drowsiness, although this change was not statistically significant. No significant differences were found in the remaining transitions. Full transition matrices and corresponding statistical comparisons are provided in the (S6).

### Power spectral density analysis

The relative Power Spectral Density (rPSD) was analyzed for each channel in each dog, then averaged across all channels, and compared between the two groups.

Fig 3A shows that in the Wakefulness state, the rPSD values for the frequency bands between [3, 10] Hz are significantly lower in the trazodone group than in the control group. However, for the higher frequency bands [13, 40] Hz the trend is opposite, with the control group showing lower rPSD values compared to the trazodone group. In the drowsinessstate (Fig 3B), the results are similarly as the Wakefulness state, but the difference in rPSD between the two groups is greater for frequencies above 13 Hz.

In the NREM state (Fig 3C), the differences in rPSD values between the two groups disappear for the lower frequency bands (below 15 Hz), except for a strength window between [0.5, 3] Hz, which remains consistent with the wakefulness and drowsiness states. For Frequencies above 13 Hz exhibit behavior similar to the wakefulness state; however, the difference between the trazodone group and the control group is significantly higher.

Finally, in the REM state (Fig 3D), the differences between the two groups are minimal, except for narrow frequency bands that do not exhibit a clear trend. It is important to note that, in this state, the number of samples in the trazodone group is 60% lower than in the control group.

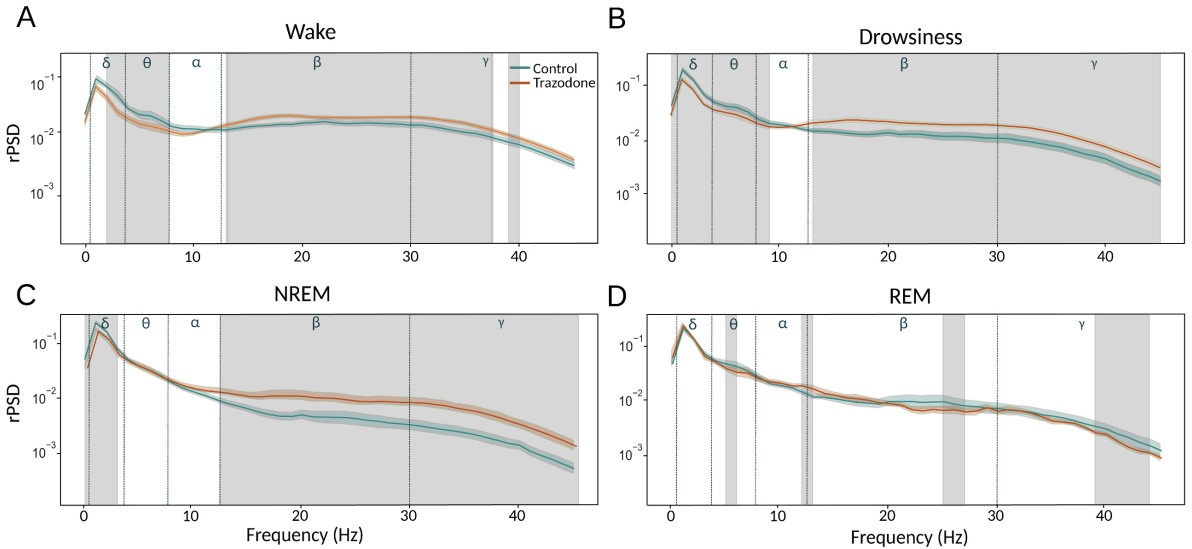

**Fig 3**. **Relative Power spectral Density (rPSD) analysis over the EEG signals.** rPSD values were averaged across all channels for the control group (green line) and compared to the same group under the influence of trazodone (red line) across the four behavioural states: **(A)** Wakefulness, **(B)** Drowsiness, **(C)** NREM, and **(D)** REM. Blue lines mark the boundaries of the electrophysiological frequency bands ($\delta$ delta, $\theta$ theta, $\alpha$ alpha, $\beta$ beta, and $\gamma$ gamma). Statistical comparisons were conducted using paired Wilcoxon signed-rank test for Wakefulness and Drowsiness and the Mann-Whitney test for NREM and REM, with significant differences ($p < 0.01$) indicated by gray shading.

## Entropy and complexity analysis

For the non-linear analysis, Lempel-Ziv complexity (LZC) and Permutation Entropy (PE) was calculated for both the low-frequency [1, 16] Hz and high-frequency [16, 45] Hz ranges and between groups. For the PE analysis we use ordinal patterns' parameter, delay $\tau = 1$ and embedding dimension of $D = 4$. In both analysis, we present the mean values over the four channels analysed.

Fig 4A left panel shows that, in the low-frequency range, the signal LZC of dogs under the influence of trazodone is significantly higher than that of the control group during wakefulness, drowsiness, and NREM states. No significant differences in LZC were found in the REM state; however, it is important to consider the difference in the number of animals between the control and trazodone groups in this state. Similar result were found using PE (Fig 4B left panel). For higher bands (Fig 4A right panel), no differences were found between the two groups for any state in the high frequency range. Furthermore, the difference in complexity between states is not significant for either group (Fig 4B right panel). All statistical data can be found in S7 and S8.

Differences in LZC and PE between sleep states within the same group were also analyzed. In the low-frequency band, both groups showed a decrease in complexity and entropy values as the dogs progressed into deeper sleep stages. Additionally, during REM sleep, there was an increase in both complexity and entropy compared to NREM in the control condition, but not in the trazodone condition. However, it is important to note that only 40% of the dogs in the trazodone group entered REM sleep. Table 2 presents the $p$-values between states for both condition, calculated using a Kruskal-Wallis test with Dunn's post-hoc correction. In the high-frequency band, no significant differences in complexity or entropy were observed between sleep states in either condition.

## Connectivity analysis

The first connectivity analysis between all channels was calculated using the Phase Lag Index (PLI) in the bands delta [2, 4] Hz, theta [5, 7] Hz, alpha [9, 11] Hz, beta [19, 21] Hz, and gamma [34, 36] Hz. Subsequently, the difference in PLI

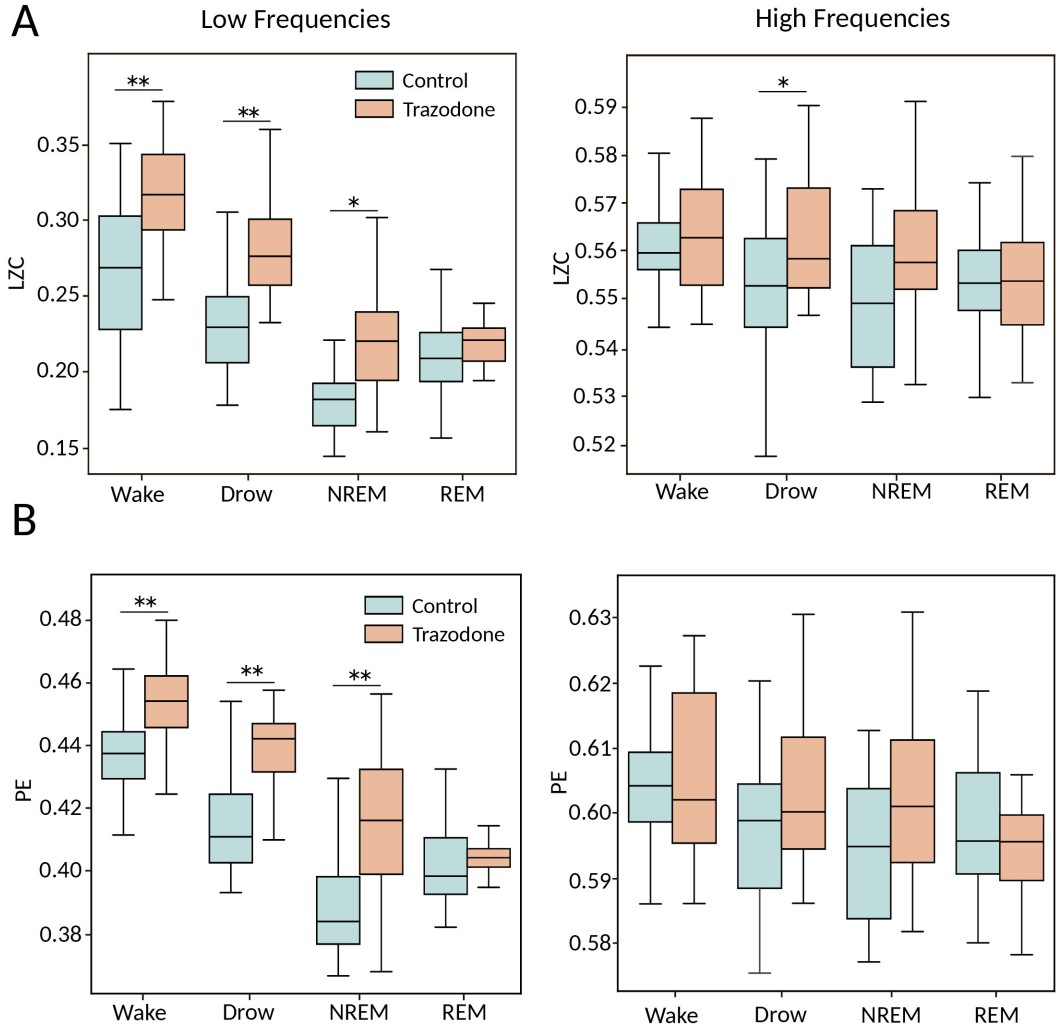

**Fig 4**. **Lempel Ziv Complexity (LZC) and Permutation Entropy (PE) analysis**. **(A)** Comparison of LZC values between dogs without (green) and under the effect of trazodone (red) across across behavioural stages. Analysis was performed for low frequency [1, 16] Hz (left panel) and high frequency [16, 45] Hz (right panel). **(B)** Similar comparison but using PE analysis with the parameter used $D = 4$ and $\tau = 1$. The boxplots show the values averaged over all four recording channels per dog. Statistical analysis between groups was performed using the Wilcoxon signed-rank test for Wakefulness and Drowsiness and Mann-Whitney test for NREM and REM (*$p < 0.05$, ** $p < 0.01$).

**Table 2**. **Statistical comparison (*p*-values) of Lempel-Ziv complexity (LZC) and Permutation Entropy (PE), for low frequency components of the signal across sleep states within each condition.** Analyses were conducted using the Kruskal-Wallis test followed by Dunn's post-hoc correction.

| States | LZC low | | PE low | |
|---|---|---|---|---|
| | Control | Trazodone | Control | Trazodone |
| Wake - Drow | $2.24\ 10^{-2}$ | $9.8\ 10^{-3}$ | $9.9\ 10^{-4}$ | $6.9\ 10^{-4}$ |
| Wake - NREM | $4.8\ 10^{-20}$ | $8.42\ 10^{-17}$ | $4\ 10^{-21}$ | $1.6\ 10^{-14}$ |
| Wake - REM | $3.95\ 10^{-6}$ | $1.83\ 10^{-10}$ | $5.5\ 10^{-9}$ | $1.5\ 10^{-13}$ |
| Drow - NREM | $6.37\ 10^{-10}$ | $2.94\ 10^{-7}$ | $2.9\ 10^{-8}$ | $2.1\ 10^{-4}$ |
| Drow - REM | - | $6\ 10^{-5}$ | - | $6\ 10^{-6}$ |
| NREM - REM | $1.7\ 10^{-3}$ | - | $6.2\ 10^{-3}$ | - |

values between the control state and the state under trazodone was calculated for each animal (for further details, please refer to the Methods ).

Fig 5A illustrates the changes in connectivity across different bands that were induced by the effect of trazodone in the four sleep states. The green links indicate an increase in connectivity under the influence of trazodone. No significant decreases in functional connectivity were observed under the drug condition for any behavioral state. The intensity of the lines represents the strength of the difference between states.

In the wakefulness state (green nodes), the Alpha band shows increased trazodone state connectivity between the central channels Cz and Fz and the right electrode F4. A comparable outcome is observed in the beta band, which also demonstrates increased interconnection between the two central channels Cz-Fz . In the gamma band, there is a high degree of connectivity between the electrodes F4-Cz and Fz-F3.

In the drowsiness state (blue nodes), there is a greater degree of connectivity between electrodes F4-Cz in the delta band for trazodone condition. In the alpha band, dogs under the drugs presents a notable enhancement in the interconnectivity between the right region at F4 and the central area, comprising Cz, Fz, and the left region, F3.

Finally, for the NREM state (red nodes), we observe increased connectivity in the theta band between channels Cz-F3 under trazodone condition. The same result is found in the alpha band. In the beta and gamma bands, there is an increase in connectivity between the central channels. A similar result is observed in the gamma band. No significant results were found for the REM state.

To complement the connectivity analysis, we examined the spectral coherence between EEG channels. Coherence analysis revealed that, across all channel pairs and in the wakefulness, drowsiness, and NREM states, coherence values under the effects of trazodone were consistently higher from 10 Hz onwards (see Figs 5B and S9, S10, S11, S12, S13, S14). However, no statistically significant differences were identified in the wakefulness state for any electrode pair.

Particularly, in the drowsiness state, significant increases in coherence were observed in interhemispheric electrode pairs, specifically F3–F4 in the [14, 25], [40, 41], and [48, 50] Hz bands (see Fig 5B). Additional significant differences were identified between Cz–Fz in the [17, 41] Hz range, and between F3, Fz in the [16, 22] and [13, 21] Hz bands.

During NREM sleep, significant differences were found only in the F3–Fz electrode pair, within the [13, 50] Hz frequency range (see Fig 5B). In the REM state, no significant alterations in coherence were observed between conditions.

## Discussion

Trazodone primarily increases serotonin levels in the brain by inhibiting its reuptake, and; it also blocks the 5-HT-2 serotonin receptors. It belongs to the class of serotonin receptor antagonists and reuptake inhibitors (SARIs). This drug also reduces the action neurotransmitters associated with arousal effects, such us noradrenaline, dopamine, acetylcholine, and histamine; low-dose trazodone use exerts a sedative effect on sleep through the antagonism of the 5-HT-2A receptor, H1 receptor, and alpha-1-adrenergic receptors [50]. Hence, trazodone is commonly used for their calming and sleep-inducing effects, with antidepressant effects at higher doses [51]. However, it is important to note that the effects of this drug can vary significantly depending on whether it is administered as a single dose or as part of a longer treatment course. Immediately following trazodone administration, initial effects like sedation or behavioral changes are often observed, which are characteristic of the drug's rapid onset of action. At low doses, in particular, trazodone primarily functions as a hypnotic [51]. Due to these sedative properties, trazodone is also used to reduce stress [30], alleviate anxiety [31,52], and support preoperative and postoperative recovery [32,53]. With prolonged use, however, the body may experience more sustained effects, such as improved sleep, alterations in brain chemistry associated with depression, or modulations in the neurotransmitter systems that trazodone influences [54].

In this study, we studied the sleep circles and analysed electroencephalogram (EEG) signals from dogs following the administration of a single dose of trazodone. The primary objective was to investigate changes in brain dynamics across wakefulness and different sleep states – drowsiness, NREM, and REM– within the same group of animals, both with and

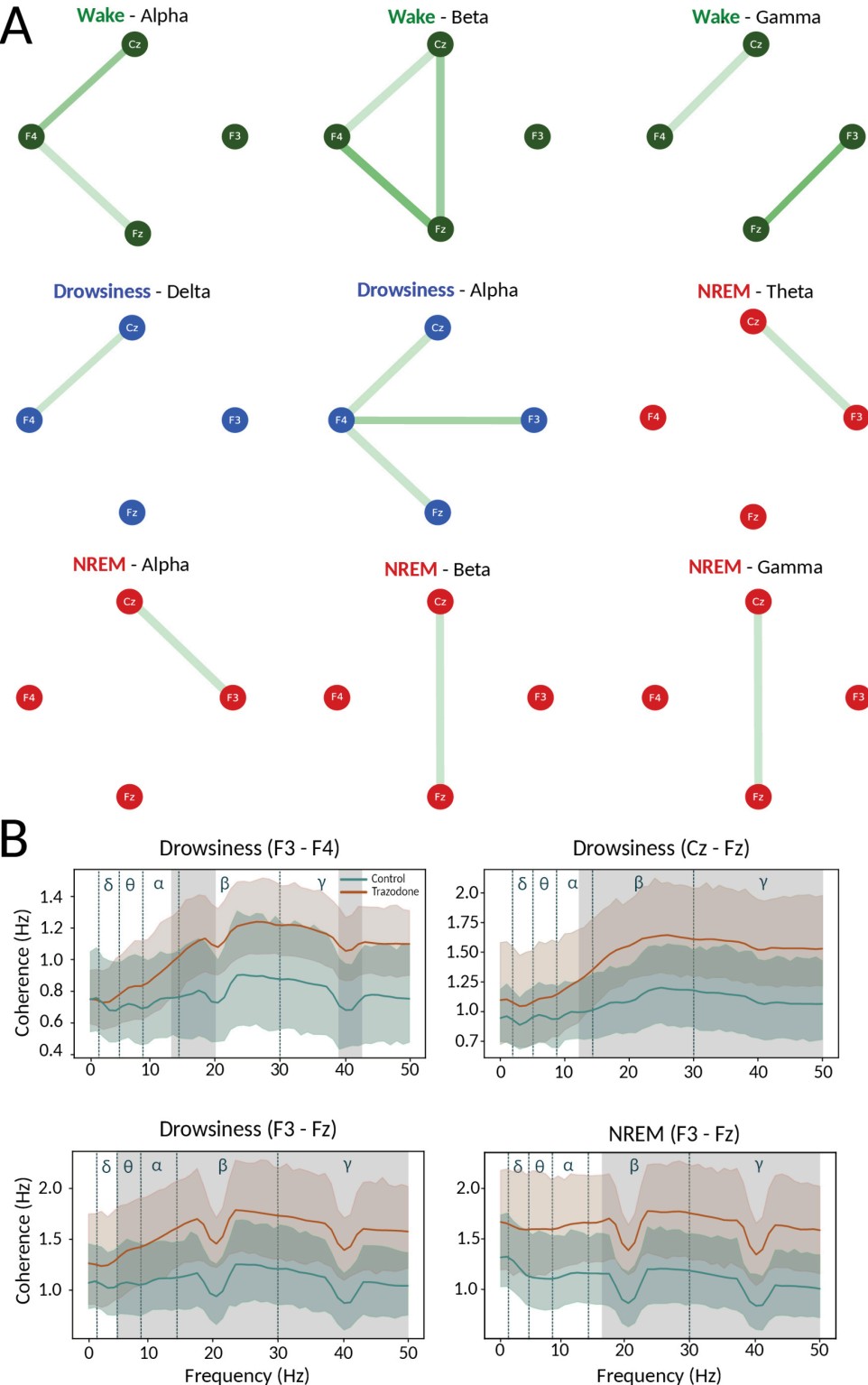

**Fig 5. Connectivity analysis for both condition. (A)** Phase Lag Index (PLI) analysis for all pair of electrodes. Green bars indicate a significant increase in phase connectivity between electrodes under the influence of trazodone. Node colors represent different sleep states: Wakefulness (green), Drowsiness (blue), and NREM (red). No significant differences were found for the REM state. **(B)** Coherence analysis focusing on the electrode pairs that exhibited the strongest effects. The green line represents the control group, while the red line represents the trazodone group. Shaded gray areas

indicate statistically significant differences between the two groups (paired Wilcoxon signed-rank test for Wakefulness and Drowsiness and the Mann-Whitney test for NREM and REM, *p* < 0.05). Blue dashed vertical lines mark the boundaries of physiological frequency bands: delta, theta, alpha, beta, and gamma. Drops around 20 Hz and 40 Hz may reflect harmonic attenuation from the notch filter.

without trazodone administration. First, we quantified the duration of time the animals spent in each sleep state and examined the dynamical changes in the waking–sleep cicle by analysing the complexity of the hypnogram and the probability of transitions between sleep stages. Subsequently, the EEG signals were subjected to a multifaceted analysis, including: (a) Power spectral analysis, to study the linear characteristics of the signals; (b) Entropy and Complexity metrics, to evaluate non-linear components of the signals; and (c) Phase lag index (PLI) and Coherence analysis, to assess changes in brain connectivity between recording electrodes.

The administration of trazodone over the time has been demonstrated to enhance the quality of sleep in humans and rats through a reduction in the proportion of time spent in a wakefulness state and an increase REM, NREM sleep [54–56]. However, in the present study, no significant alterations in the time spent by the dogs in the wakefulness and NREM states were observed under the influence of trazodone. This may be attributed to the lack of prolonged treatment. An increase in time spent in the drowsiness stage was observed, though it did not reach statistical significance. It is also notable that 60% of the dogs did not enter the REM stage under the influence of trazodone. This finding is consistent with the results of studies on the use of trazodone in depressed patients, which have demonstrated that those treated with the drug over a one-week period exhibited suppressed REM sleep, increased state 1 NREM, and a reduction in stage 2 NREM [54,57]. In addition, we observed significant changes in wake-sleep cycle following the administration of the drug. The prominent decrease in Lempel–Ziv complexity in animals treated with trazodone suggests a reduction in the variability of the wake-sleep cycle. These findings indicate a more structured and less fragmented sleep pattern under trazodone, likely reflecting the drug's sedative and stabilizing effects on neural activity. Similarly, the analysis of transition probabilities between sleep states suggests that, while trazodone promotes prolonged drowsiness, it may disrupt or diminish the stability of REM sleep. The observed reduction in the probability of transitioning between sleep states under trazodone further highlights a disruption in the natural wake-sleep cycle.

In the relative Power Spectral Density analysis, we observed clear differences between the trazodone and control groups in different wakefulness and sleep states. In the waking state, the rPSD in the lower frequency bands ([3, 10] Hz) was significantly reduced in the trazodone group compared to the control group. Conversely, an opposite trend was observed in the higher frequency bands ([13, 45] Hz), with the trazodone group showing higher rPSD values than the control group. Both for the low and high frequency EEG bands, the alteration in the neurotransmitter dynamics, mainly that form part of the activating system [58], must be affected by trazodone. A similar pattern was observed during drowsiness, reinforcing the idea that trazodone has a consistent effect on brain activity across different states. In support of these findings, studies in rats with low doses of trazodone (2.5 mg/kg) [54] showed a decrease in rPSD in the range of [6, 16] Hz during the wakefulness state within the first hour of administration. This decrease is consistent with our observations. However, it's important to note that the rat study did not report rPSD results during sleep, which limits direct comparisons. During NREM sleep, the relative Power spectral density exhibited a pattern similar to that observed during wakefulness and drowsiness, with lower rPSD values in the trazodone group, particularly in the delta and portions of the theta frequency bands. This reduction in low-frequency activity may suggest that trazodone modulates the synchronization of slow-wave activity across cortical areas—an essential component of restorative sleep processes [59]. Conversely, in higher frequency bands (above 13 Hz), dogs treated with trazodone showed significantly increased rPSD compared to the control group, indicating that the drug may enhance or alter cortical activation during NREM sleep. In the REM sleep state, no significant differences were found between groups. However, due to the considerably smaller number of animals in the trazodone group compared to the control group, no definitive conclusions can be drawn at this stage.

The analysis of complexity and entropy revealed a notable increase in complexity in animals under trazodone during the wakefulness, drowsiness, and NREM states in the lower frequency bands ([1 16] Hz). This indicates that trazodone may induce more irregular or unpredictable patterns in the brain's electrical activity at these lower frequencies. No significant differences were identified between the groups for any of the states in the higher frequency bands ([16 45] Hz), suggesting that the impact of trazodone on complexity and entropy may be frequency-dependent. Furthermore, it was observed that as the animals progressed from wakefulness towards drowsiness and NREM sleep, that there was a reduction in complexity and entropy in the lower frequency bands, a pattern that was evident in both groups. A similar outcome was observed in a study conducted on rats [41,60].

A comparison of these findings with the rPSD analysis reveals an inverse relationship between rPSD and the measures of LZC and PE. In the rPSD analysis, for the wakefulness, drowsiness, and NREM states, the lower frequency band rPSD was reduced in the trazodone group. This inverse relationship is to be expected, given that rPSD measures the periodic components of EEG signals, whereas LZC and PE assess the non-linearity or aperiodicity of the signal. Based on these observations, it appears that the administration of trazodone results in a reduction in energy within the low-frequency components of the signal, accompanied by an increase in complexity. This may be attributed to the possibility that a reduction in the power spectrum may indicate a desynchronization in the synaptic potentials received by the neural groups measured by the electrode. As the degree of synchronization among neural networks declines, the variability in local field potentials increases, which could result in a raise of complexity.

In the higher frequency bands (above 13 Hz), as the dogs progress to deeper sleep states, those treated with trazodone demonstrate enhanced synchronization in the postsynaptic potentials of neurons, resulting in a higher power spectrum. However, no significant change in LZC and PE was observed between the two groups. This suggests that while trazodone enhances synchronization at higher frequencies, it does not necessarily alter the non-linear dynamics of these signals, as reflected in the measures of LZC and PE.

A study of connectivity between the recorded electrodes revealed a general increase in connectivity—both in phase and power—across all wakefulness and sleep states in the group under the influence of trazodone. However, this increase was specific to certain electrode pairs, depending on the behavioural state and the frequency band under analysis. These findings indicate that trazodone modulates brain connectivity in a state-dependent and frequency-specific manner, potentially enhancing communication within specific neural networks. The observed increases in connectivity may reflect the impact of trazodone on the underlying neurophysiological processes that govern synaptic efficacy and network synchronization. This modulation of connectivity may contribute to the broader changes in brain function and sleep architecture induced by trazodone, highlighting its complex effects on neural dynamics during different stages of sleep.

While this study offers valuable insights into the acute effects of trazodone on brain dynamics and sleep architecture, several limitations must be acknowledged. First, the use of a single acute dose in a dog model may not fully capture the complexity of chronic administration, which better reflects clinical use. The acute effects observed here may therefore differ from those emerging under prolonged treatment, where adaptations in brain chemistry and sleep patterns could yield distinct outcomes. Moreover, the scarcity of prior studies using single-dose trazodone in animals constrains our ability to place these findings in the broader literature, underscoring the need for additional work in this domain. Second, daytime sleep propensity is known to vary across dogs as a function of age, breed, and individual characteristics [21,61–63]. Although our within-subject design reduced the impact of this variability on trazodone vs. control comparisons, such differences may still have contributed to inter-individual variability in baseline measures. Third, although our cohort included dogs of varying breeds, ages, and body sizes—factors known to influence longevity, sleep propensity, and physiological function—the within-subject design minimized potential confounding by ensuring that trazodone and control recordings were compared within the same individual. Consistent with this, we observed no significant correlations between body weight, age, and key sleep parameters such as REM latency (S4). Nevertheless, breed- and size-related factors may influence sleep and EEG dynamics in larger or more heterogeneous populations, and future studies are

warranted to explore these effects more systematically. Finally, differences in skull thickness and head size can alter absolute EEG power through attenuation and spatial smoothing. We mitigated this by employing a standardized montage and impedance thresholds, and by relying on metrics relatively insensitive to amplitude scaling (LZC, PE, PLI/coherence). For PSD analysis, we used normalized power spectra to standardize all signals. From a statistical standpoint, paired non-parametric tests were used for wakefulness and drowsiness, while unpaired tests were required for REM and NREM due to missing data. We note that paired designs assume symmetry of differences and reduce usable sample size, which should be considered when interpreting these results.

## Conclusion

This study provides a comprehensive analysis of the effects of trazodone on canine brain dynamics and sleep architecture. The findings of this study demonstrate that only one dose of trazodone induces notable alterations in sleep structure and electroencephalogram (EEG) activity, particularly through a reduction in power within lower frequency bands, an increase in complexity, and an enhancement of connectivity within specific neural networks. These effects are most pronounced during NREM sleep, indicating that trazodone promotes a more intricate and more synchronised neural state. This research highlights the significance of elucidating the short-term impacts of trazodone on brain function, offering novel insights into its broader neurological effects beyond its primary clinical applications.

## Supporting information

**S1 Table. Mean $\pm$ standard deviation of the number of clean and noisy epochs across all dogs.** The third column reports the percentage of artifact-free (good) epochs in each state relative to the total number of recorded epochs. (PDF)

**S2 Table. Number of clean and noisy epochs, and the percentage of clean epochs from total epochs per dog analyzed in this study.** (PDF)

**S3 Table. Time spent (min) in each behavioral state for every dog under control and trazodone conditions.** (PDF)

**S4 Table. Latency to each sleep state (min) for individual dogs under control and trazodone conditions.** (PDF)

**S5 Fig. Correlation between REM latency with (A) Age and (B) Weight.** (TIFF)

**S6 Fig. Transition matrices between different sleep states for Control and Trazodone conditions (left and center, respectively).** Asterisks indicate statistically significant differences. *p*-values represent comparisons between matrices in both conditions. Statistical analysis was performed using the Kruskal–Wallis test followed by Dunn's post hoc correction. (TIFF)

**S7 Table. Statistical significance of Lempel-Ziv Complexity differences in low and high frequency bands between experimental conditions.** *p*–values were calculated using Wilcoxon signed-rank tests for paired comparisons Wakefulness (Wake) and Drowsiness (Drow), and Wilcoxon rank-sum tests for unpaired comparisons (NREM vs. REM). n/s: not significant ($p > 0.05$). (PDF)

**S8 Table. Statistical significance of Permutation Entropy differences in low and high frequency bands between experimental conditions.** *p*–values were calculated using Wilcoxon signed-rank tests for paired comparisons Wakefulness (Wake) and Drowsiness (Drow), and Wilcoxon rank-sum tests for unpaired comparisons (NREM vs. REM). n/s: not significant ($p > 0.05$).
(PDF)

**S9 Fig. Coherence analysis for Cz-Fz electrode pairs across all behavioral states.** The green and red traces denote the control and trazodone groups, respectively. Vertical blue dashed lines indicate boundaries of standard physiological frequency bands (delta $\delta$, tetha $\theta$, alpha $\alpha$, beta $\beta$, gamma $\gamma$). Observed attenuations near 20 Hz and 40 Hz likely reflect harmonic suppression by the notch filter.
(TIFF)

**S10 Fig. Similarly to S9, for F3-FCz electrode pairs.**
(TIFF)

**S11 Fig. Similarly to S9, for F3-F4 electrode pairs.**
(TIFF)

**S12 Fig. Similarly to S9, for F3-Fz electrode pairs.**
(TIFF)

**S13 Fig. Similarly to S9, but for F4-Cz electrode pairs.**
(TIFF)

**S14 Fig. Similarly to S9, but F4-Fz electrode pairs.**
(TIFF)

## Author contributions

**Conceptualization:** Alejandra Mondino, Pablo Torterolo, Hugo Aimar, Natasha J. Olby, Diego Martin Mateos.

**Data curation:** Magaly Catanzariti, Alejandra Mondino.

**Formal analysis:** Magaly Catanzariti, Diego Martin Mateos.

**Funding acquisition:** Pablo Torterolo, Hugo Aimar, Natasha J. Olby.

**Investigation:** Natasha J. Olby, Diego Martin Mateos.

**Methodology:** Alejandra Mondino, Pablo Torterolo, Diego Martin Mateos.

**Project administration:** Natasha J. Olby, Diego Martin Mateos.

**Software:** Magaly Catanzariti.

**Supervision:** Pablo Torterolo, Diego Martin Mateos.

**Validation:** Alejandra Mondino, Natasha J. Olby, Diego Martin Mateos.

**Visualization:** Magaly Catanzariti, Diego Martin Mateos.

**Writing – original draft:** Diego Martin Mateos.

**Writing – review & editing:** Alejandra Mondino, Pablo Torterolo, Hugo Aimar, Natasha J. Olby, Diego Martin Mateos.

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
