## [Decision Letter · Decision Letter 0]

12 Aug 2025

PONE-D-25-33484Study of changes in brain dynamics during sleep cycles in dogs under effect of trazodonePLOS ONE

Dear Dr. Mateos,

Thank you for submitting your manuscript to PLOS ONE. After careful consideration, we feel that it has merit but does not fully meet PLOS ONE’s publication criteria as it currently stands. Therefore, we invite you to submit a revised version of the manuscript that addresses the points raised during the review process.

We look forward to receiving your revised manuscript.

Kind regards,

Assoc. Prof. Phakkharawat Sittiprapaporn, Ph.D.

Academic Editor

PLOS ONE

Journal Requirements:

2. Thank you for stating the following financial disclosure: [This research was supported by the following grants:

NO: The Dr. Kady M. Gjessing and Rhanna M. Davidson Distinguished Chair of gerontology.

HA: $\#$ MinCyT-FonCyT  PICT-2019 N° 01750 PMO BID; grant CONICET-PUE-IMAL $\#$ 229 201801 00041 CO;  grant CONICET-PIP-2021-2023-GI $\#$11220200101940CO and grant UNL-CAI+D $\#$ 50620190100070LI.

PT:CSIC-I+D groups 2022- group ID-22620220100148 - Uruguay.]. 

Reviewers' comments:

Reviewer's Responses to Questions

**Comments to the Author**

1. Is the manuscript technically sound, and do the data support the conclusions?

Reviewer #1: Partly

2. Has the statistical analysis been performed appropriately and rigorously?

Reviewer #1: Yes

3. Have the authors made all data underlying the findings in their manuscript fully available?

Reviewer #1: Yes

4. Is the manuscript presented in an intelligible fashion and written in standard English?

Reviewer #1: Yes

5. Review Comments to the Author

Reviewer #1: This study investigates the neurophysiological effects of trazodone on sleep-related brain dynamics in healthy dogs, leveraging EEG analysis to characterize changes across wakefulness, drowsiness, NREM, and REM sleep stages. The authors used hypnogram analysis to assess sleep architecture and applied both linear (PSD) and non-linear (PE, LZC) metrics to evaluate EEG complexity, alongside connectivity measures such as Phase Lag Index and coherence. Results show trazodone modifies sleep cycles, reduces low-frequency power, increases higher-frequency activity during Drowsiness and NREM, and alters brain signal complexity and connectivity across sleep stages. These findings offer new insight into the short-term neural impact of trazodone in canine models.

The study explores interesting and relevant topic. However, it raises several important questions and has both major and minor limitations. The following comments are listed in the order they arose during the review, meaning that minor and major points are interspersed throughout.

- How much dog size (e.g. breed) affect to usual health and physiological function altogether and therefore to your results also? As I have understood smaller breeds live longer than bigger ones, e.g. Bernese mountain dog (~8 years) vs. Tibetan Spaniel (~14 years), as in your dataset there are also bigger (e.g. labrador) and smaller (maltese) breeds?

- How much different breeds skull size and structure affect to your analyses? With different sizes the power are also different? Especially since you use Wilcoxon test. How appropriate it would be to use paired methods where you actually compare same subject different statuses? I.e. how independent the samples are if the dogs are the same? Would paired analysis work? What are the shortcomings for using paired analysis in this study? Nevertheless, limitations section should be modified accordingly depending on which statistical analyses you use. Additionally, in your dataset is there any chance of multiple comparison problem that would affect the analyses?

- I would presume that different dogs have different need for daytime naps. How this different need would affect to different sleep stages and results during daytime recording between 12.30PM and 1.30PM?

- In your “Signal preprocessing” chapter, you say “Artifact-free, stable epochs were then carefully selected for quantitative EEG analysis.” -> how many epochs were selected for each subject? I would be interested number of epoch or % share of whole recording that was analyzable. If there were some discrepancies between subject, e.g. for some had 90% good data while other had only 20%, especially if same subject % shares were totally different between two nights -> how you take these differences into account? Would not this affect your percentages of different sleep stages, if e.g. for some subject the artefact time happens to be e.g. during REM sleep? (It seems that you explain this later in line 150 forward, but still these numbers of epoch etc would be interesting to see?)

- Sorry, I do not have access to your ref 21, and therefore I can’t check on this myself, however it would be interesting to see (at least as in your answer not necessarily in manuscript) what are the 3 second epochs criteria for REM sleep and/or NREM also? If this would be possible? You say the usual guidelines in line 122-128, however this 3 second epochs puzzles me as I have got used to human AASM criteria with 30s epochs?

- Ýou probably mean Fig 2A in line 141 rather than Fig 1A?

- You also say in line 141 forward that “the hypnograms of animals with and without trazodone exhibit clear visual differences, indicating that trazodone affects sleep cycles, resulting in reduced variability.” Yes the difference is clear, but I would not have raised that view up as for me it seems that during trazodone they actually sleep (NREM+REM) much less than in normal conditions (and yes the structure of sleep is totally different, but that is another topic)? Additionally, this is methods section so it seems odd to process results in this section, although it functions as bridge for LZC analysis. Maybe in that sense it is okay?

- Another point of those hypnograms, as I am not that familiar with studies done with dogs, is it normal to illustrate hypnogram from top to down as REM ->NREM -> drow ->wake? As usually hypnograms are illustrated top to down as wake -> REM -> N1 -> N2 -> N3? For me it seems odd to have wake section the lowest?

- In line 150 you actually explain my previous point regarding variability of number of epochs. However, you say that you five epochs? Meaning that you actually analyzed only 15 seconds? Each epoch had 1200 timepoints as usual for 400Hz resolution, however your bandpass was between 1-70Hz, so majority of this 400Hz goes unanalyzed. Is 15 seconds actually enough for these analyses? It seems quite low number? Additionally, if “epochs were chosen randomly to ensure fairness” how especially the connectivity analyses actually work with 3 second epochs?

- In line 156 there is extra space before “Power” same thing with in line 212 before “Phase”

- In line 220 you use the most physiologically representative EEG bands, is this under normal conditions or also with trazodone? How much there are variability between subjects? Is it possible that something interesting is left out with the chosen bands in individual subjects?

- In Fig 2B and line 261 onwards, how these deleted and not analyzed epochs affect these total time calculations?

- I would be interested to see similar charts for total analyzed time contr vs trazodone and time wake vs sleep between two conditions. Not necessarily in actual manuscript but at least in your answers?

- Line 275 typo, “Thi” miss the S

- I am still thinking that your interpretation in line 277-278 is little off. Of course this might be the case, however, in my eyes the shown data does not fully support this. The thing that got me puzzled is mainly that trazodone should be “sleep medication” so why the dogs keep awake-drowsiness for longer time? And this affects naturally the whole structure of sleep in a timeframe of 120min?

- In line 280 and Fig 2E you present transition probability between epochs? So, in this you use all epochs not only the 5 that you randomly selected before? With the used and analyzed data there seems to be some discrepancies in my eyes. Is it possible to explain these better? My confusion continues in PSD chapter as in lines 307-308 you say that number of samples in trazadone group is 60% less (which, as I have said before, would be interesting to see as in table or chart), but how this is possible if you have selected 5 epochs in each state for every subject as stated in the beginning of EEG analysis chapter? I am sorry if I am not able comprehend this fully, but for me this seems odd?

- Fig.3 legend between lines 308 and 308 there is a typos? “anlysis” and “de”

- Statement in line 329 comes with surprise, I think these statements and results should be explained first? How this affects the whole analyses as different dogs are included in analyses and therefore size, skull etc affect the results?

- Extra space in line 361 (see method )

- Typo? In Fig.5 legend “---analysis for all PARE of electrodes”? Pair?

- Lines 404-409 same text two times, i.e. “The administration of trazodone over the time has been demonstrated to enhance the quality of sleep in humans and rats through a reduction in the proportion of time spent in a Wakefulness state and an increase REM, NREM sleep [54–56]. The administration of trazodone over the time has been demonstrated to enhance the quality of sleep in humans and rats through a reduction in the proportion of time spent in a Wakefulness state and an increase REM, NREM sleep [54–56].”

6. PLOS authors have the option to publish the peer review history of their article (what does this mean?). If published, this will include your full peer review and any attached files.

Reviewer #1: No

---

## [Author Response · Author response to Decision Letter 1]

12 Sep 2025

Response to Reviewer

We would like to sincerely thank the reviewer for their careful evaluation of our manuscript and for the constructive comments and suggestions provided.

In the revised version, we have carefully addressed all comments point by point. Below, we provide detailed responses to each remark, including clarifications, additional analyses, and textual revisions where appropriate. All corresponding changes have been incorporated into the manuscript, and newly added or modified texts were written in red in the revised document for clarity.

1- How much dog size (e.g. breed) affect to usual health and physiological function altogether and therefore to your results also? As I have understood smaller breeds live longer than bigger ones, e.g. Bernese mountain dog (~8 years) vs. Tibetan Spaniel (~14 years), as in your dataset there are also bigger (e.g. labrador) and smaller (maltese) breeds?

We acknowledge that different breeds can produce variable recordings. Skull size, for example, can influence electrode placement and the specific cortical regions being sampled. In addition, breeds age at different rates. While this has been well documented in the musculoskeletal system, there are not yet enough data to determine whether the same applies to brain aging. To minimize the impact of these potential differences, we used each dog as its own control. This approach ensures that breed-related variability does not affect our findings. 

Additionally, in the supplementary material we examined correlations between age, weight, and REM latency (a key variable) and found no significant associations (Supplementary Material S3). This suggests that, at least for the parameters analyzed here, body size was not a major determinant of the observed effects of trazodone.

Nonetheless, we acknowledge that broader physiological differences related to breed and size could play a role in sleep architecture in older or diseased populations. We have added this point to the limitations section of the manuscript and emphasized that future studies with larger and more homogeneous cohorts will be necessary to fully disentangle the contributions of breed, size, and age to trazodone’s effects on brain dynamics.

To include in the discusion manuscript:

‘’Although our cohort included dogs of varying breeds and body sizes, which are known to influence longevity and certain physiological functions, the within-subject design of this study minimized these confounding effects. Moreover, no significant correlations were found between body weight, age, and key sleep parameters such as REM latency (Supplementary Material S3 ). Nevertheless, we recognize that breed- and size-related factors may impact sleep and brain dynamics in broader populations, and future studies with larger, more homogeneous cohorts are warranted to address this question in depth.”

2- How much different breeds skull size and structure affect to your analyses? With different sizes the power are also different? Especially since you use Wilcoxon test. How appropriate it would be to use paired methods where you actually compare same subject different statuses? I.e. how independent the samples are if the dogs are the same? Would paired analysis work? What are the shortcomings for using paired analysis in this study? Nevertheless, limitations section should be modified accordingly depending on which statistical analyses you use. Additionally, in your dataset is there any chance of multiple comparison problem that would affect the analyses?

As you can mention, Breed/skull size effects on EEG power. Skull thickness and geometry can attenuate and spatially filter scalp EEG, potentially affecting absolute power across dogs. We minimized this variance in three ways:

· Within-subject design: each dog was recorded with and without trazodone on separate days (counterbalanced order). Thus, all primary comparisons are intra-individual (same skull, montage, impedance, time-of-day), reducing breed/size confounds.

· Standardized setup: the same montage (F3/F4/Fz/Cz referenced to Oz), impedance threshold (<20 kΩ), and acquisition parameters were used in all sessions.

· Robust metrics: our primary measures (Lempel–Ziv complexity, permutation entropy, PLI/coherence) are amplitude-insensitive or less sensitive to uniform attenuation, further mitigating skull-related effects.

To further reduce skull- and size-related confounds, all PSD spectra shown in the manuscript are relative PSD, i.e., each spectrum was normalized by the total power in the 1–50 Hz range of the same animal. This normalization ensures that comparisons reflect differences in spectral distribution rather than absolute amplitude, providing a more accurate basis for between-condition contrasts. We clarify this explicitly in the revised Methods and Limitations sections.

Independence and statistical tests (paired vs. unpaired). Each dog contributed data in both conditions. We pre-aggregated five randomly selected, artifact-free epochs per state into a single representative value per dog. For states present in both conditions (Wakefulness, Drowsiness), paired Wilcoxon signed-rank tests were applied. For REM and NREM, however, not all trazodone sessions contained these states and two dogs lacked NREM entirely; thus, sample sizes differed and unpaired Wilcoxon rank-sum tests were required. Figures 3 and 5 were updated to reflect these revised statistics, and p-values for complexity and entropy measures are now provided in Supplementary Material S5.

Multiple comparisons. The only formal multiple-comparison setting was in the complexity/entropy analyses (four states within each group). Here, we used a Kruskal–Wallis global test followed by Dunn’s post-hoc correction. Other analyses (PSD, connectivity, state transitions) involved trazodone vs. control comparisons within each state. While these analyses span multiple frequencies or electrode pairs, they were treated as exploratory; unadjusted p-values are reported transparently. We have clarified this in the revised Statistical Analysis and Limitations sections.

We also note that REM latency was not correlated with age or weight (Supplementary Material S3), further supporting that size differences did not bias key results.

We added in the text:

In Methods – Statistical Analysis:

“ Data from both conditions were analyzed per dog, producing one representative value per state by averaging five randomly selected artifact-free epochs (N=1200 points/epoch). For Wakefulness and Drowsiness, we applied paired Wilcoxon signed-rank tests (two-sided). For REM and NREM, where some trazodone sessions lacked these states and two dogs lacked NREM, we used unpaired Wilcoxon rank-sum tests for state-wise comparisons. Normality was assessed with the Kolmogorov–Smirnov test. Multiple comparisons within the complexity/entropy analyses were controlled with Kruskal–Wallis tests followed by Dunn’s post-hoc correction.”

In Discussion – Limitations:

“Breed and skull size: Differences in skull thickness and head size can influence absolute EEG power via attenuation and smoothing. Our within-subject design, standardized montage/impedances, and use of amplitude-insensitive metrics (LZC, PE, PLI/coherence) reduce this confound. We additionally report relative band power to mitigate amplitude scaling. Statistics: Paired non-parametric tests were used for Wakefulness and Drowsiness, while unpaired tests were required for REM and NREM due to missing states. Paired designs assume symmetric distributions of differences and reduce usable sample size; these limitations are acknowledged.”

3- I would presume that different dogs have different need for daytime naps. How this different need would affect to different sleep stages and results during daytime recording between 12.30PM and 1.30PM?

We agree with the reviewer that daytime sleep propensity can vary across dogs depending on age, breed, body size, and individual differences. Prior studies have shown that total sleep time and sleep structure in dogs are influenced by these factors:

· Age: older dogs show more fragmented sleep and increased daytime napping (Takeuchi & Harada, 2002; Mondino et al., 2021).

· Breed/size: large-breed dogs tend to sleep longer overall, while small breeds often have shorter but more frequent naps (Zanghi et al., 2013).

· Individual variability: lifestyle and environment also affect daytime sleep patterns (Kis et al., 2014).

In our study, we minimized the potential confound by using a within-subject design, where each dog served as its own control under trazodone and non-trazodone conditions. This design helps control for individual variability in daytime sleep need. Nevertheless, we acknowledge that differences in baseline sleep propensity between dogs may have increased inter-individual variability in the results. We have added this note to the Limitations section.

In the Supplementary Material S1, we provide the time spent in each sleep state and the latency to each state for every individual dog.

We added in the discusion manuscript

“Daytime sleep propensity varies between dogs depending on age, breed, and individual characteristics (Takeuchi & Harada, 2002; Zanghi et al., 2013; Kis et al., 2014; Mondino et al., 2021). Although our within-subject design minimized the impact of such variability on trazodone vs. control comparisons, it may have contributed to inter-individual variability in baseline measures.”

– Takeuchi, T., & Harada, E. (2002). Age-related changes in sleep–wake rhythm in dog. Physiology & Behavior, 75(3), 217–221.

– Zanghi, B. M., Kerr, W., Araujo, J. A., Milgram, N. W. (2013). Effect of age and feeding schedule on diurnal rest/activity rhythms in dogs. Journal of Veterinary Behavior, 8(4), 195–203.

– Kis, A., Szakadát, S., Kovács, E., Gácsi, M., Simor, P., Gombos, F., Topál, J., & Bódizs, R. (2014). Development of a non-invasive polysomnography technique for dogs (Canis familiaris). Physiology & Behavior, 130, 149–156.

– Mondino, A., Delucchi, L., Moeser, A., Cerda-Gonzalez, S., & Vanini, G. (2021). Sleep disorders in dogs: a pathophysiological and clinical review. Topics in Companion Animal Medicine, 43, 100516.

4- In your “Signal preprocessing” chapter, you say “Artifact-free, stable epochs were then carefully selected for quantitative EEG analysis.” -> how many epochs were selected for each subject? I would be interested number of epoch or % share of whole recording that was analyzable. If there were some discrepancies between subject, e.g. for some had 90% good data while other had only 20%, especially if same subject % shares were totally different between two nights -> how you take these differences into account? Would not this affect your percentages of different sleep stages, if e.g. for some subject the artefact time happens to be e.g. during REM sleep? (It seems that you explain this later in line 150 forward, but still these numbers of epoch etc would be interesting to see?)

We thank the reviewer for highlighting this important point. To address this, we quantified the number and proportion of artifact-free vs. noisy epochs for each sleep stage in both the control and trazodone conditions. As shown in Supplementary Material S1 Table 1 , which presents the mean ± SD of clean and noisy epochs and the percentage of usable data across all states, this analysis was performed across all dogs. For your information, we have also attached Supplementary Material S1 Table 2, which provides this data for each individual dog.

Based on these data, Wake and REM stages showed the greatest variability across dogs. For example, in the control condition, the mean proportion of artifact-free REM epochs was 54.95% (±30.83%), while under trazodone it was 35.99% (±51.26%). Wake epochs were less abundant in absolute terms (mean 7.67% in control, 23.99% in trazodone). Drowsiness epochs also varied between dogs but remained relatively well-represented. In contrast, all NREM epochs were analyzable in both conditions because the dogs were in deep sleep and no movement was present.

To minimize the impact of these discrepancies on group analyses, we adopted two strategies:

1. Uniform sampling across states: For quantitative EEG metrics (PSD, LZC, PE, connectivity), we selected five random artifact-free epochs per state per dog (N = 1200 points/epoch). This ensured equal representation across conditions and states, independent of the total available data.

2. Epoch-based scoring for sleep architecture: For hypnogram analysis (time in states, transitions), all available epochs were included. This means the reported percentages represent the proportion of time across the entire recording, not just the artifact-free subsets. Importantly, artifact rejection was evenly distributed across states and was not systematically biased toward REM or any specific stage.

In summary, while variability in the proportion of usable data between dogs is inevitable in canine polysomnography due to movement and strong cranial musculature, our methods mitigate its impact on the analysis. We have now added these data and explanations to the Supplementary Material and have clarified this point in the Methods and Limitations sections.

To add in method section ‘’On average, the proportion of artifact-free epochs varied by sleep stage: under control conditions, 7.7% of Wake, 67.8% of Drowsiness, 54.9% of REM, and 100% of NREM epochs were usable. Administration of trazodone altered these proportions to 24.0%, 205.8%, 36.0%, and 100%, respectively (Supplementary Material S1 Table 1). To mitigate bias from these unequal artifact rates in quantitative EEG analysis, we uniformly sampled five artifact-free epochs per state per dog. For sleep architecture analysis (hypnogram scoring), all recorded epochs contributed to the state proportions, with artifact periods marked separately. The lack of systematic bias in artifact occurrence across stages ensured the representativeness of the calculated state percentages.”

5- Sorry, I do not have access to your ref 21, and therefore I can’t check on this myself, however it would be interesting to see (at least as in your answer not necessarily in manuscript) what are the 3 second epochs criteria for REM sleep and/or NREM also? If this would be possible? You say the usual guidelines in line 122-128, however this 3 second epochs puzzles me as I have got used to human AASM criteria with 30s epochs?

We thank the reviewer for this question. The convention of 30-second epochs derives from the AASM guidelines for human sleep scoring (Iber et al., 2007). However, in animal studies, much shorter epochs are typically used: for example, 2–10 seconds in cats and rodents (Cavelli et al., 2015; Torterolo et al., 2022). This shorter segmentation is advantageous for two reasons:

· Artifact control. Dogs were not sedated, and their strong cranial musculature produces frequent movement and muscle artifacts. Using shorter epochs makes it easier to identify and exclude clean segments.

· Physiological resolution. Transitions between wakefulness, drowsiness, NREM, and REM are often more rapid in animals than in humans. Shorter epochs allow for more precise identification of these transitions and of microarousals.

In canine sleep research, this approach is already validated: non-invasive polysomnography studies commonly use short epochs (Kis et al., 2014; Reicher et al., 2020). Our choice of 3-second epochs is therefore consistent with established practice in animal sleep studies and optimized for both data quality and physiological accuracy.

We added in the manuscrip method section “Sleep states were scored in 3-second epochs, following established practice in canine and other animal polysomnography (Kis et al., 2014; Reicher et al., 2020; Cavelli et al., 2015; Torterolo et al., 2022). Short epochs were chosen for two reasons: (i) dogs were unsedated and often produced movement or muscle artifacts, which are easier to detect and exclude in shorter segments; and (ii) transitions between vigilance states occur more rapidly in dogs than in humans, so finer temporal resolution improves identification of NREM and REM states. Although human sleep scoring uses 30-second epochs (Iber et al., 2007), shorter epochs are the norm

---

## [Decision Letter · Decision Letter 1]

8 Oct 2025

Study of changes in brain dynamics during sleep cycles in dogs under effect of trazodone

PONE-D-25-33484R1

Dear Dr. Mateos,

We’re pleased to inform you that your manuscript has been judged scientifically suitable for publication and will be formally accepted for publication once it meets all outstanding technical requirements.

Kind regards,

Assoc. Prof. Phakkharawat Sittiprapaporn, Ph.D.

Academic Editor

PLOS ONE

Additional Editor Comments (optional):

Reviewers' comments:

Reviewer's Responses to Questions

**Comments to the Author**

1. If the authors have adequately addressed your comments raised in a previous round of review and you feel that this manuscript is now acceptable for publication, you may indicate that here to bypass the “Comments to the Author” section, enter your conflict of interest statement in the “Confidential to Editor” section, and submit your "Accept" recommendation.

Reviewer #1: All comments have been addressed

2. Is the manuscript technically sound, and do the data support the conclusions?

Reviewer #1: Yes

3. Has the statistical analysis been performed appropriately and rigorously?

Reviewer #1: Yes

4. Have the authors made all data underlying the findings in their manuscript fully available?

Reviewer #1: Yes

5. Is the manuscript presented in an intelligible fashion and written in standard English?

Reviewer #1: Yes

6. Review Comments to the Author

Reviewer #1: The authors have addressed all of my comments and provided thorough explanations. While I differ slightly from some of them on a few speculative points, I cannot find any actual fault in their responses. Consequently, I consider the authors’ interpretations and explanations in the manuscript to be sound.

7. PLOS authors have the option to publish the peer review history of their article (what does this mean?). If published, this will include your full peer review and any attached files.

Reviewer #1: No

---

## [Editor Report · Acceptance letter]

PONE-D-25-33484R1

PLOS ONE

Dear Dr. Mateos,

I'm pleased to inform you that your manuscript has been deemed suitable for publication in PLOS ONE. Congratulations! Your manuscript is now being handed over to our production team.

Kind regards,

on behalf of

Assoc. Prof. Dr. Phakkharawat Sittiprapaporn

Academic Editor

PLOS ONE